# APT-Pipe: An Automatic Prompt-Tuning Tool for Social Computing Data Annotation

## ABSTRACT

Recent research has highlighted the potential of LLM applications, like ChatGPT, for performing label annotation on social computing text. However, it is already well known that performance hinges on the quality of the input prompts. To address this, there has been a flurry of research into *prompt tuning* — techniques and guidelines that attempt to improve the quality of prompts. Yet these largely rely on manual effort and prior knowledge of the dataset being annotated. To address this limitation, we propose APT-Pipe, an automated prompt-tuning pipeline. APT-Pipe aims to automatically tune prompts to enhance ChatGPT's text classification performance on any given dataset. We implement APT-Pipe and test it across twelve distinct text classification datasets. We find that prompts tuned by APT-Pipe help ChatGPT achieve higher weighted F1-score on nine out of twelve experimented datasets, with an improvement of 7.01% on average. We further highlight APT-Pipe's flexibility as a framework by showing how it can be extended to support additional tuning mechanisms.

## KEYWORDS

Human computation, crowdsourcing, prompt-tuning, large language models, data annotation

**ACM Reference Format:**
Anonymous Author(s). 2018. APT-Pipe: An Automatic Prompt-Tuning Tool for Social Computing Data Annotation. In *Proceedings of Make sure to enter the correct conference title from your rights confirmation emai (Conference acronym 'XX)*. ACM, New York, NY, USA, 11 pages. https://doi.org/XXXXXXX.XXXXXXX

## 1 INTRODUCTION

Data annotation commonly relies on human intelligence and suffers from challenges related to annotator quality, data comprehension, domain knowledge, and time cost [9, 20, 25, 53]. Recent research has highlighted that prompt-based LLM applications, like ChatGPT, have the potential to substitute human intelligence as a data annotator for social computing tasks, *e.g.,* sentiment analysis [4, 64] and hate speech detection [26]. Use of tools like ChatGPT for such annotation tasks can significantly improve the cost and speed for researchers. However, such models are heavily impacted by prompt quality, whereby poorly tuned prompt can severely damage annotation accuracy [52]. This creates the need for tools that can help researchers design and evaluate effective prompts.

To date, most prompt-tuning for annotation tasks is done either manually [35] or by injecting prior knowledge of the dataset [16, 26]. Prompt-tuning for ChatGPT, however, usually introduces demands on expertise that can be hard to address. Thus, we argue that an automated prompt-tuning tool for ChatGPT tasks could remove a significant manual load related to learning prior knowledge of the dataset to annotate. Researchers have already begun to propose techniques for such automation [4, 40, 46]. We focus on three early-stage ideas: introducing standardized templates that (almost) guarantee consistent responses; introducing few-shot examples into prompts to guide annotations; and embedding additional text metadata to augment raw text information. In practice, however, implementing these simple concepts raises key challenges. **First**, ChatGPT can generate unpredictable and unstructured responses that make parsing annotations difficult (as its responses lack a standard format). Although cloze-style prompts can improve responses' parsability, such templates usually introduce a trade-off on ChatGPT's time efficiency [46]. **Second**, although preliminary evidence suggests that performance can be improved by including few-shot examples of annotation [40], manual exemplar selection can be time consuming. Thus, techniques need to be developed to automatically identify suitable few-shot examples for the prompt. **Third**, although embedding natural language processing (NLP) metrics in prompts [4] has been shown to increase performance, the selection of such metrics requires domain specific expertise. Thus, techniques need to be developed to select the best permutations of NLP metrics to include in prompts. Importantly, the suitability of the above approaches may differ on a per-dataset basis, making a single standardized approach impractical.

To address these challenges, we propose the Automatic Prompt-Tuning Pipeline (*APT-Pipe*), an extensible prompt-tuning framework for improving text-based annotation task performance. APT-Pipe only requires humans to annotate a small sample subset of data to offer a robust ground truth. It then exploits this annotated subset to automatically tune and test prompts with the goal of improving ChatGPT's classification performance. APT-Pipe addresses the above three challenges using a three-step prompt-tuning pipeline. It first generates JSON template prompts that can make ChatGPT respond consistently in a desired format. APT-Pipe then automatically computes the most suitable few-shot exemplars to include in prompts. Finally, through a number of iterations, APT-Pipe tests a variety of prompt configurations with NLP metrics to identify which work best for the particular dataset. These three steps are derived from our experience and state-of-the-literature. However, we acknowledge that many other prompt-tuning techniques are likely to emerge in the coming years. Consequently, we design APT-Pipe in an extensible and modular fashion, allowing other researchers to easily "plug-in" additional tuning strategies.

We implement APT-Pipe with the ChatGPT `gpt-3.5-turbo` model and evaluate it across twelve distinct text classification datasets drawn from three domains in social computing. APT-Pipe will be made publicly available for use. Our results shows that prompts tuned by APT-Pipe can improve ChatGPT's classification performance on *nine out of twelve* datasets, with a 7.01% increase of

*Conference acronym 'XX, June 03–05, 2018, Woodstock, NY*
2018. ACM ISBN 978-x-xxxx-xxxx-x/YY/MM. . . $15.00
https://doi.org/XXXXXXX.XXXXXXX

weighted F1-score on average. Moreover, prompts tuned by APT-Pipe can decrease ChatGPT's computing time for classification, while providing over 97% of all responses in a consistent and easy-to-parse format. Finally, we highlight APT-Pipe's flexibility by extending its framework with two state-of-art tuning mechanisms called Chain-of-Thought and Tree-of-Thought.

The contributions of this paper can be summarized as follows:

- We propose APT-Pipe, an automatic and extensible prompt-tuning pipeline to improve ChatGPT's performance for social computing data annotation.
- We implement and validate APT-Pipe's effectiveness. We show that APT-Pipe can improve ChatGPT's classification performance on nine out of twelve different datasets, with improvement on ChatGPT's F1-score by 7.01% on average.
- We show that APT-Pipe prompts generate annotations in a consistent format and take less time compared to existing prompt designs. On average, 97.08% responses generated by APT-Pipe are parsable by desired format and it decreases more than 25% time cost of the baselines.
- We highlight APT-Pipe's extensibility by introducing two additional prompt-tuning techniques, Chain of Thought (CoT) and Tree of Thought (ToT). We show that CoT improves APT-Pipe's F1-scores on five datasets by 4.01%, and ToT improves APT-Pipe's F1-scores on six datasets by 3.49%. Importantly, we show that APT-Pipe can learn these improvements on a per-dataset basis, automatically tuning the prompts without human intervention.

## 2 BACKGROUND & RELATED WORK

In this section, we present a review of the literature on LLMs' applications in social computing and prompt-tuning for text annotation.

### 2.1 LLMs in Social Computing

Social computing research relies extensively on human-annotated datasets. Such annotations are time consuming and often costly. The release of LLM tools such as ChatGPT [4, 18] has uncovered a range of possibilities to automate text data labeling tasks due to their robust performance in natural language comprehension and reasoning [30].

Recently, some studies [26, 49] have employed LLMs to assist in text annotation in social computing. Sallam et al. [49] employ LLMs to discriminate misinformation about vaccines on social media platforms. Huang et al. [26] exploit LLMs to annotate implicit hateful tweets and report nearly 80% accuracy. Zhu et al. [64] evaluate the potential of LLMs to annotate five different types of text data. However, work has revealed that the annotation performance of LLMs varies across different LLMs and different domains and datasets [1, 50]. The above literature demonstrates that, although it is feasible to employ LLMs for text annotation tasks, the generality and reliability of LLMs need to be improved.

### 2.2 Prompt-Tuning for Text Annotation

In existing LLM-based text annotation methods [2, 33, 37, 48], several prompt-tuning techniques have been utilized to enhance the performance of LLMs. Alizadeh et al. [2] investigate the performance difference between LLMs and crowd-workers in text annotation tasks. They employ natural language formatted prompts

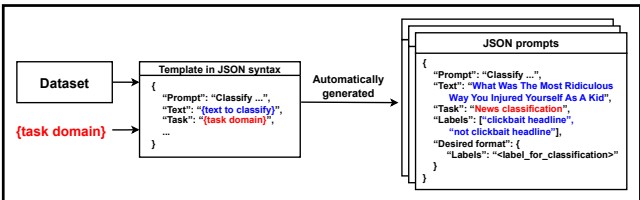

**Figure 1: Flow chart of Step 1. Prompt on the right-hand side shows example text for clickbait news headline detection.**

in zero-shot tests. For few-shot tests, they employ the Chain-of-Thought (CoT) [54] prompting technique, where LLMs are provided with the question and step-by-step reasoning answers as examples. Kuzman et al. [33] conduct genre identification tasks using LLMs by feeding prompts with manually defined task descriptions, and manually extracting the answers from responses. Reiss [48] applies LLMs to text annotation tasks by manually designing 10 prompts, feeding them to LLMs, and evaluating their performance. From the above literature, we observe that existing prompt-tuning methods require the participation of domain expertise and human judgement, which hinders the utilization of LLMs in automated annotation tasks. Using ChatGPT as a case-study, we focus on proposing a pipeline that can automatically perform prompt tuning.

## 3 APT-PIPE DESIGN

We now describe our pipeline for tuning a text classification prompt. The pipeline's goal is to generate a tuned prompt to improve Chat-GPT's classification performance on a per-dataset basis. The pipeline operates in discrete steps, and we have designed our framework in an extensible fashion, allowing future modifications to be "plugged-in" as new steps. The current implementation has three steps. First, the prompt engineer must personalize a JSON-based prompt following our template (Step 1). Following this, the pipeline searches for suitable few-shot learning examples to include in the prompt (Step 2) and augmentary text metrics that can guide ChatGPT's reasoning (Step 3). To inform this decision, APT-Pipe iteratively tests prompt variations on a small subset of manually annotated data (typically a few hundred samples) to learn the best combination of information to include in the final prompt. Once complete, the prompt can be used on the remaining dataset.

### 3.1 Step 1: Preparing Initial Prompt

Figure 1 illustrates the process to prepare the initial prompts. The prompt engineer must first input an initial prompt using a JSON syntax. We use JSON because encoded notations can facilitate language models' ability to understand and respond with more precise answers [59, 63]. Moreover, designing prompts as JSONs is convenient for prompt augmentation, as APT-Pipe can easily embed additional information using suitable key-value pairs [32, 59]. Note, we have also experimented with various commonly applied prompt templates, *i.e.* natural language [4], dictionary [64], cloze question [24], and JSON [65]. Based on our findings, we observe that JSON prompts generate more precise responses in a consistent format. Although the engineer could input any preferred JSON template, by default, we rely on the following template:

```
{
    "Prompt": "Classify the following text by given labels for specified
    task.",
    "Text": "{text to classify}",
    "Task": "{task domain}",
    "Labels": ["Label 1", "Label 2", ...],
    "Desired format": {
        "Label": "<label_for_classification>"
    }
}
```

This prompt template must then be populated with values by the user. The initial sentence clarifies that the prompt pertains to text classification, with "Text" serving as a placeholder for the input text that the user wishes to annotate. Naturally, each individual text item will be annotated one-by-one using a separate prompt invocation. "Task" specifies the domain for classification (*e.g.,* stance detection), and "Labels" enumerates the potential labels for classifying the text. The user must fill these out to reflect the specific annotation task specification. Finally, the "Desired Format" specifies the format that ChatGPT should utilize to form its answers. Afterwards, the generated classification prompts in Step 1 are treated as inputs for the pipeline to conduct prompt-tuning in Step 2 next.

### 3.2  Step 2: Prompt-tuning with Few-shot

In this step, the pipeline tunes the prompt taken from Step 1 by employing a *few-shot* learning approach. This setting gives the model a few demonstrations of the correct annotations at inference time. Such a technique enables the model to study "in-context" knowledge from given examples and provide better decisions [7]. We embed this into the existing JSON prompt text by including a set of samples from the dataset alongside their ground-truth labels. The challenge here is identifying the best samples to include in the few-shot examples. Figure 2 presents an overview and we detail the process as follows.

APT-Pipe first conducts exemplar selection. To select the samples to include from the manually annotated set, we rely on the method proposed by Yang *et al.* [56]. Specifically, we select the text samples from the dataset with the highest textual similarity with the text to classify. Thus, we use OpenAI's `text-embedding-ada-002` [44] model to generate the embeddings for each data sample in the dataset and then calculate the pairwise cosine similarity between the embedding of each exemplar and the embedding of the text to classify. Consequently, we focus on the top $n$ exemplars with the highest cosine similarity. Note, $n$ is configurable, allowing the user to select a range of potential values to experiment on. By default, $n = 5$ as a five-shot setting is commonly used in related works [19, 38]. We include the $n$ examples in the template as follows:

```
"Examples": [
    {"Text": "{exemplar 1}", "Label": "{exemplar 1's ground-truth
    label}"},
    {"Text": "{exemplar 2}", "Label": "{exemplar 2's ground-truth
    label}"},...]
```

APT-Pipe then uses the small data subset, manually annotated by the prompt engineer, to test if the few shot additions improve performance. First, APT-Pipe requests ChatGPT to annotate the subset without any few-shot examples included; this serves as the baseline.

APT-Pipe then repeats the process, including the $n$ few-shot examples (note, $n$ can cover a range of values). For all configurations, APT-Pipe calculates the weighted F1-score of the classification, by comparing ChatGPT's outputs to the human ground truth labels. We do this check because prior work has shown that including few-shot examples can actually degrade performance on certain datasets [45, 62]. If the F1-score does not drop, APT-Pipe outputs the prompt that attained the highest F1 score improvements. If the F1-score drops, we remove the key-value pairs of the examples and then pass the original prompt to Step 3.

### 3.3  Step 3: Prompt-tuning with NLP Metrics

As highlighted in [24, 58], domain-relevant text knowledge can play a helpful role in prompt-tuning. Therefore, in Step 3, APT-Pipe tunes the prompt by augmenting it with additional metadata that describes the text to annotate. Figure 3 presents an overview, as well as a pseudo-code representation in Algorithm 1. For this, APT-Pipe first calculates a pre-defined set of NLP metrics for all items of the dataset. Our implementation currently includes sentiment, emotion, toxicity, and topic metrics. We emphasize that our pipeline is extensible and can include any other metadata.

To prepare for Step 3, we initially divide the data annotated by the engineer into two segments: a training set comprising 75% of the data, and a validation set consisting of the remaining 25%. Following this, we begin an iterative process.

**Step 3.1:** First, APT-Pipe takes the labeled dataset that was annotated by ChatGPT using the previous Step 2 prompt (without any additional metrics included in the prompt). This offers a baseline to compare results for other prompts that include metrics.

**Step 3.2:** Following this, it is necessary to select which metric(s) will be included in the test prompt. To achieve this, we first calculate the metrics for every item in the small manually annotated dataset. For example, if there are four metrics, then each row in the data will be supplemented with four new columns. One option would be to then generate prompts containing every combination of metrics, and test every one to select the best. However, this would be prohibitively slow and expensive. Thus, it is necessary to estimate which metrics are most likely to result in a performance improvement in a more lightweight fashion.

To achieve this, we train an XGBoost classifier (detailed in § 4.5) to estimate the predictive power of each metric. Specifically, we construct a new dataset that contains each data sample, the ground-truth human label, the ChatGPT label (from Step 3.1), and each of the metrics. We then train the classifier to predict if the label generated by ChatGPT will be *identical* to the original human annotated label, using only ChatGPT's label, the raw text, and its additional metric (*i.e.* this is a binary task — either the two labels match, or they do not). Note, we test metrics individually — thus, each metric will have a separately trained classifier. We then calculate the F1 score of each classifier using 10-fold cross-validation, and rank the metrics based on their scores. Our intuition is that metrics that better predict ChatGPT's ability to generate the correct labels will be more suitable for inclusion in the prompt. This step allows us to reduce the search space and avoid trying every combination of metrics.

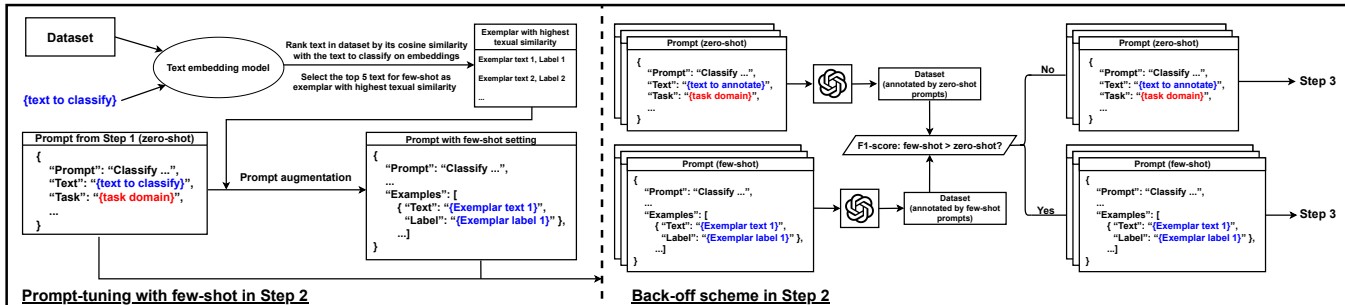

**Figure 2: Flow chart of Step 2.**

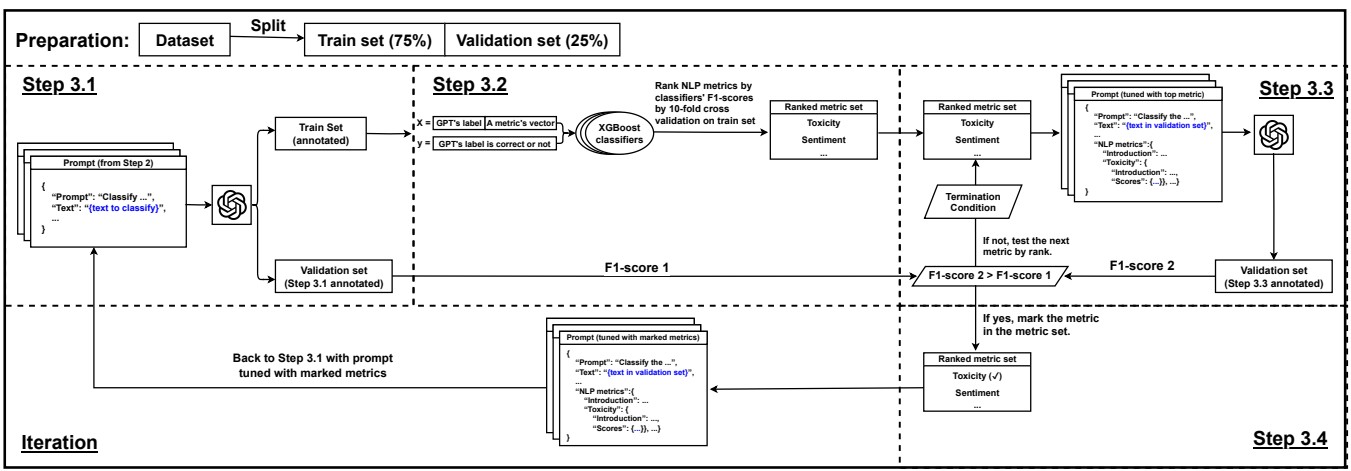

**Figure 3: Flow chart of Step 3.**

**Step 3.3:** Next, we select the top-ranked metric and add it to the existing prompt, according to the template below. Using this new test prompt, we ask ChatGPT to annotate the separate validation set and record its proposed labels. This allows us to study if the addition of the metric improves ChatGPT's performance.

```
"NLP metrics": {
    "Introduction": "Refer to the following NLP metrics of
    the text to make classification.",
    "{metric (e.g., Sentiment)}": {
        {corresponding key-value pairs (detailed in §4.3)}
    },...}
```

**Step 3.4:** If the inclusion of the metric shows a weighted F1-score improvement over the Step 2 baseline, we mark that metric for inclusion in the final prompt.

**Iteration:** APT-Pipe then repeats the iteration. We go back to Step 3.1 and re-label the training data using the newly created prompt (which includes the metric marked in Step 3.4 from the previous iteration). For example, if the prior iteration has shown that the inclusion of the toxicity metric improves performance, it is included in the prompt on the next iteration. This re-labeled training data is then used to train again the XGBoost metric classifiers, and select the next metric to add into a new prompt (Step 3.2). Note, only

the remaining metrics will be considered, and the ranking of their importance may differ from the previous iteration. Steps 3.3 and 3.4 are then repeated to test if the new metric enhanced ChatGPT's performance. Note, at this stage, the newly created prompt will include the highest ranked metric, plus all prior metrics that have been shown to improve performance.

**Termination Condition:** The loop will continue until all metrics in Step 3.3 have been tested. Thus, the number of iterations is limited by the number of metrics. Importantly, by ranking metrics using XGBoost, we avoid having to experiment with every permutation. Although this cannot be guaranteed to bring optimum performance for all datasets, our experiments have shown that this is an effective estimate. Once all metrics have been tested, APT-Pipe will return a final prompt containing all the chosen metrics as JSON attributes. This becomes the fine-tuned prompt that the prompt engineer will use.

## 4 EXPERIMENTAL SETTINGS

In this section, we illustrate the selection of social computing datasets for experiment, evaluation metrics, and our pipeline implementation.

## 4.1 Task Domains and Datasets

Our experiments cover twelve public datasets drawn from three significant domains as highlighted by reviews on text classification [11, 31, 41]:

- **News classification:** *Ag's News* [61], *SemEval-19* [29], *Sarcasm* news headline dataset [42], and *Clickbait* news headline dataset [8].
- **Stance detection:** *SemEval-16* [43], *P-Stance* [34], *Vax-Stance* [47], and *RU-Stance* [64].
- **AI-writing detection:** *Tweepfake* [15], *FakeHate* [12, 22], *GPT-Article* [39], and *GPT-Wiki* [23].

We summarize these datasets' statistics and descriptions in Table 7. For any dataset with more than 3,000 items, we subset it to 3,000 items using stratified sampling by its classification labels. We do this due to the rate and cost limitations of the ChatGPT API.

## 4.2 Evaluation Metrics

We assess the tuned prompts' performance by our pipeline based on the following evaluation metrics:

- **Classifier performance:** We view ChatGPT as a text classification engine and use standard classifier metrics: weighted F1-score, precision, and recall.
- **Responses' parsability:** The percentage of parsable responses based on the desired format. We assume a response is *parsable* only if it reports the label in the desired format. We use regular expressions to match a string in JSON syntax from ChatGPT's response and then parse it into a JSON object.
- **Time cost:** The time it takes for ChatGPT to return an annotation response. We normalize the request time by comparing it to the request time of a null prompt sent before. Here, the normalization is done to remove the influence of the network transmission. Afterwords, we divide the normalized request time by the number of words in the response. This metric can be interpreted as the time required for ChatGPT to generate each token.

## 4.3 NLP Metrics for Prompt Augmentation

In Step 3, we automatically select metadata metrics to include in the prompt. For the purpose of our evaluation, we experiment with four examples of NLP metrics. Table 1 summarizes how we include the metrics in each prompt. The metrics are:

- **Sentiment:** This metric indicates the text's sentiment polarity. We use a pre-trained sentiment analysis model called *XLM-T* to measure the sentimental polarity of the text using a three-dimensional vector represented as positive, negative, and neutral [5]. For prompt augmentation, we input sentimental polarity (ranging from 0 to 1) based on above three dimensions.
- **Emotion:** This metric indicates the text's emotional leaning. We utilize a well-known model called *Emotion English DistilRoBERTa-base*, which uses a seven-dimensional vector, covering Ekman's six basic emotions with a neutral class [14, 21]. For prompt augmentation, we input emotional leaning (ranging from 0 to 1) for the above seven listed emotions.
- **Toxicity:** This metric indicates the percentage of toxic content contained in the text. We use the *Google Perspective API* to measure the existence of toxic content. We construct a six-dimensional vector covering the six Perspective attributes [28]

| | Prompt Augmentation |
|---|---|
| **Sentiment** | "Introduction": "Scores of sentiment leaning of text (ranging from 0 to 1).", "Scores": {"Positive": `0.xx`, "Neutral": `0.xx`, "Negative": `0.xx`} |
| **Emotion** | "Introduction": "Scores of emotion leaning of text (ranging from 0 to 1).", "Scores": {"Anger": `0.xx`, "Disgust": `0.xx`, "Fear": `0.xx`, "Joy": `0.xx`, "Neutral": `0.xx`, "Sadness": `0.xx`, "Surprise": `0.xx`} |
| **Toxicity** | "Introduction": "Scores of toxicity degree of text (ranging from 0 to 1).", "Scores": {"Overall Toxicity": `0.xx`, "Severe Toxicity": `0.xx`, "Identity Attack": `0.xx`, "Insult": `0.xx`, "Profanity": `0.xx`, "Threat": `0.xx`} |
| **Topic** | "Introduction": "Representative words to describe the major topic of the text.", "Words": [`keywords to describe text's major topic by BERTopic`] |

**Table 1: A summary of key-value pairs augmented in the prompt text for prompt-tuning on NLP metrics. The blue text denotes the probability or text output by the corresponding model.**

used to measure toxicity (*e.g., TOXICITY, IDENTITY_ATTACK*). For prompt augmentation, we input the API's scores (ranging from 0 to 1) for these six attributes.

- **Topic:** This metric indicates the main topics of the text. Given a text, we train a *BERTopic* model using the dataset it belongs to and then infer its topic embeddings according to the trained model [17]. For prompt augmentation, we extract the most representative topic as the one with the highest possibility in its BERTopic embedding. We then input into the prompt the list of important words of this topic reported by the model. Note, alternative configurations could include a larger number of topics.

## 4.4 Baselines

We select several basline prompts generated by existing tuned templates to compare with the APT-Pipe generated prompts.

***Cloze prompt:*** This template designs prompts as a cloze-style question. It asks ChatGPT to replace the blank or masked part of the text. Prior studies have utilized cloze prompts to improve ChatGPT's performance on text classification [19, 55]. We generalize a cloze prompt template as follows:

```
Fill [Label] for {task domain} task with a label in [Label 1, Label 2,
...].
The text "{text to classify}" is classified as [Label].
Desired format:
Label: <label_for_classification>
```

***Dictionary prompt:*** This template is a generalized pattern representing prompts formed as a dictionary. These are commonly applied in text classification studies on ChatGPT [4, 13, 64]. We generalize a dictionary prompt as follows:

```
Classify the following text by given labels for specified task.
Text: "{text to classify}".
Task: {task domain}.
Labels: [Label 1, Label 2, ...].
Desired format:
Label: <label_for_classification>
```

***JSON prompt:*** This template represents the prompts encoded in JSON format in Step 1, without any further prompt-tuning (§ 3.1).

***Parsing responses:*** When each baseline prompt generates a response, it is necessary to parse it and extract the returned annotation label. For responses generated by the dictionary and cloze prompts, we index ChatGPT's classification label baseline as the first word of the substring after "label:". For responses generated by JSON prompts, we parse them into a JSON object and extract ChatGPT's classification label using the key "label"

## 4.5 Experimental Implementation

***ChatGPT setting:*** We have built the full APT-Pipe tool, and integrated it with gpt-3.5-turbo. We request access ChatGPT through OpenAI's API with parameter *temperature* set to 0 to make the response focused and offer deterministic.[1] APT-Pipe will be made publicly available on github.[2]

***Dataset Split:*** The experiment use a data split strategy where we divide each dataset into training, validation, and test sets, allocated at a ratio of 60%, 20%, and 20%, respectively. Initially, in Step 1, we merge the training and validation sets to create a single input dataset for APT-Pipe (covering 80%). Subsequently, in Step 3, APT-Pipe separates them again and uses the test and validation sets separately for the prompt-tuning with NLP metrics. Thus, we work on the assumption that users of APT-Pipe will be able to annotate a relatively small number of annotations to drive APT-Pipe's tuning. Note that the remaining 20% test set is not used within APT-Pipe and instead is used only for evaluation purposes in Section 5.

***Classifier for metric ranking:*** In Step 3, we train a classifier for each NLP metric for ranking and selecting the metrics for prompt augmentation. For this, we use Extreme Gradient Boosting (XG-Boost) [10]. We implement it using the xgboost python package, with parameter setting (*objective*="binary:logistic", *seed*=42).

## 5 RESULTS

We next evaluate APT-Pipe's ability to improve prompt performance. We further perform an ablation study to show the effectiveness of its stepwise prompt-tuning methods.

## 5.1 Overall Annotation Performance

We first compare APT-Pipe's performance against the three baselines. Table 2 presents ChatGPT's weighted F1-score, precision, and recall on the test sets of all twelve datasets. Overall, APT-Pipe achieves the highest weighted F1-score on *nine* datasets. Compared with the baseline with the second-best F1-score, APT-Pipe improves the F1-score by 7.01% ($SD = 9.74\%$) on average. This evidences that APT-Pipe is effective in improving ChatGPT's overall annotation performance. On average, APT-Pipe also improves ChatGPT's

---

[1]https://platform.openai.com/docs/api-reference/chat

[2]Redacted due to anonymous submission requirements.

weighted precision by 1.18% ($SD = 1.36\%$) for *eight* datasets, and by 4.88% ($SD = 7.70\%$) for the weighted recall rate for *nine* datasets. This indicates that APT-Pipe also enables ChatGPT to respond with accurate annotations with fewer false positive answers.

Recall that each task (news, stance and AI-writing classification) has four evaluation datasets. APT-Pipe performs the best on 3/4 of their datasets for each task. Thus, we confirm that APT-Pipe can benefit ChatGPT on annotating social computing data for news classification, stance detection, and AI-writing detection. However, the improvements in ChatGPT's performance vary across these three tasks. For news classification and stance detection, APT-Pipe offers an average improvement of 2.20% ($SD = 0.80\%$) and 1.22% ($SD = 0.48\%$) F1-score, respectively. APT-Pipe's improvements are far greater for AI-writing detection though, with an average improvement of 17.61% F1-score ($SD = 11.19\%$). Such results suggest that, although APT-Pipe can enhance ChatGPT's annotation on social computing data, its effectiveness depends on the task domain. In our case, APT-Pipe has a much larger advantage in AI-writing detection than news classification and stance detection.

## 5.2 Effectiveness of ChatGPT's Responses

As implied in relevant studies, two challenges for applying prompt-tuning for data annotation tasks are that (***i***) ChatGPT may generate responses in arbitrary formats [3, 18]; and (***ii***) tuning methods could increase the time cost for generating annotations [51]. We next assess whether APT-Pipe also faces these two difficulties.

***Parsability:*** Table 3 shows responses' parsability for ChatGPT on all 12 dataset. On average, 97.08% ($SD = 2.79\%$) of responses generated by APT-Pipe are parsable. When compared with baselines, APT-Pipe improves upon dictionary prompts, with an improvement of 16.47% parsability on average. However, the parsability is slightly reduced compared to the cloze or plain JSON prompts. Applying APT-Pipe decreases the responses' parsability by 1.83% when compared to cloze prompts, and 1.94% when compared to JSON prompts on average. These results show that applying APT-Pipe would pay a small price of responses' parsability. We conjecture this is because adding key-value pairs for prompt-tuning also increases the entropy of the prompt text, which can lead LLMs to respond to answers with a higher uncertainty [6, 60].

***Time Cost:*** Table 3 also presents the time taken for ChatGPT to return an annotation. We see that APT-Pipe improves upon both cloze and dictionary prompts. Applying APT-Pipe decreases the time cost by 0.157 seconds per token when compared to cloze prompts, and 0.082 seconds per token when compared to plain JSON prompts. However, compared with JSON prompts, APT-Pipe introduces a small additional time cost (0.041 seconds per token). These results show that the tuned prompts take slightly longer to execute compared to the un-tuned JSON. This is because the tuned prompts contain additional information, which is used to achieve performance improvement. Nonetheless, APT-Pipe is still able to boost ChatGPT's speed for data annotation compared to the dictionary and cloze prompts utilized in other literature.

## 5.3 Ablation Study

We next conduct an ablation study to measure the improvements attained by each step described in Section 3. Note, Table 2 already

| | Shot | NLP Metrics (in selection order) | F1-score | | | | Precision | | | | Recall | | | |
|---|---|---|---|---|---|---|---|---|---|---|---|---|---|---|
| | | | Cloze | Dictionary | JSON | APT-Pipe | Cloze | Dictionary | JSON | APT-Pipe | Cloze | Dictionary | JSON | APT-Pipe |
| Ag's News 🖒 | Few | Topic, Emotion, Toxicity. | 81.98% | 85.33% | 81.68% | **88.32%** | 84.85% | 85.74% | 85.06% | **88.72%** | 82.45% | 85.50% | 82.29% | **88.31%** |
| SemEval-19 🖒 | Zero | Sentiment, Topic. | 68.98% | 77.88% | 78.07% | **80.29%** | 69.86% | 78.04% | 78.01% | **81.08%** | 68.55% | 77.78% | 78.40% | **80.00%** |
| Sarcasm | Zero | Emotion. | 61.24% | **62.46%** | 52.34% | 55.31% | 67.95% | **68.35%** | 65.40% | 66.88% | 63.57% | **64.47%** | 58.01% | 59.87% |
| Clickbait 🖒 | Zero | Topic, Emotion. | 73.54% | 91.20% | 93.95% | **95.35%** | 82.74% | 91.20% | 94.29% | **94.64%** | 75.20% | 91.21% | 93.95% | **94.57%** |
| SemEval-16 🖒 | Few | Topic, Sentiment. | 49.37% | 33.79% | 37.30% | **50.93%** | 59.67% | **61.85%** | 50.51% | 57.95% | 47.00% | 29.18% | 34.33% | **47.28%** |
| P-Stance | Zero | - | 79.69% | 79.10% | 77.86% | 77.86% | 80.61% | 80.74% | **81.02%** | **81.02%** | **79.86%** | 79.28% | 78.41% | 78.41% |
| Vax-Stance 🖒 | Zero | Topic. | 64.19% | 42.27% | 65.58% | **66.25%** | 66.87% | 48.52% | 69.10% | **69.97%** | 63.25% | 46.67% | 64.53% | **64.91%** |
| RU-Stance 🖒 | Zero | Emotion. | 67.37% | 57.40% | 73.47% | **74.89%** | 80.50% | 74.40% | 81.13% | **81.58%** | 67.37% | 58.36% | 73.05% | **74.55%** |
| Tweepfake 🖒 | Few | - | 61.63% | 59.52% | 63.00% | **75.27%** | 76.07% | 66.52% | 70.83% | **76.20%** | 65.42% | 62.63% | 65.37% | **75.46%** |
| FakeHate 🖒 | Few | - | 10.49% | 19.04% | 11.64% | **49.51%** | 81.15% | 78.14% | 86.97% | 74.56% | 25.38% | 22.33% | 19.36% | 49.18% |
| GPT-Article | Few | Sentiment, Emotion, Toxicity | **60.40%** | 46.87% | 54.05% | 40.96% | **61.21%** | 50.93% | 58.72% | 48.92% | **60.78%** | 52.63% | 56.78% | 49.75% |
| GPT-Wiki 🖒 | Few | - | 55.03% | 51.62% | 46.54% | **65.11%** | 55.33% | 66.49% | 47.91% | **67.17%** | 54.83% | 61.29% | 52.94% | **64.15%** |

**Table 2: Comparison between ChatGPT's performance with baseline prompts and APT-Pipe prompts. F1-score, precision, and recall are all weighted by labels. A "🖒" denotes that APT-Pipe gains a higher F1-score than the baselines. The bold percentage denotes the highest value of the corresponding metrics.**

| | Responses' parsability | | | | Time cost | | | |
|---|---|---|---|---|---|---|---|---|
| | Cloze | Dictionary | JSON | APT-Pipe | Cloze | Dictionary | JSON | APT-Pipe |
| Ag's News | 98.98% | 57.69% | 99.83% | 99.83% | 0.439 | 0.546 | 0.197 | 0.250 |
| SemEval-19 | 100.00% | 93.60% | 100.00% | 100.00% | 0.246 | 0.272 | 0.154 | 0.168 |
| Sarcasm | 100.00% | 98.29% | 100.00% | 100.00% | 0.264 | 0.304 | 0.169 | 0.172 |
| Clickbait | 100.00% | 66.89% | 100.00% | 99.83% | 0.231 | 0.279 | 0.160 | 0.199 |
| SemEval-16 | 89.30% | 65.01% | 93.01% | 94.68% | 0.287 | 0.253 | 0.156 | 0.162 |
| P-Stance | 98.63% | 56.53% | 96.42% | 96.42% | 0.247 | 0.175 | 0.168 | 0.168 |
| Vax-Stance | 100.00% | 100.00% | 100.00% | 100.00% | 0.360 | 0.492 | 0.196 | 0.146 |
| RU-Stance | 100.00% | 97.57% | 98.95% | 97.55% | 0.280 | 0.374 | 0.172 | 0.176 |
| Tweepfake | 100.00% | 81.99% | 100.00% | 96.76% | 0.468 | 0.112 | 0.199 | 0.191 |
| FakeHate | 100.00% | 81.57% | 100.00% | 91.80% | 0.546 | 0.123 | 0.203 | 0.227 |
| GPT-Article | 100.00% | 68.21% | 100.00% | 98.53% | 0.478 | 0.079 | 0.199 | 0.263 |
| GPT-Wiki | 100.00% | 100.00% | 100.00% | 92.98% | 0.782 | 0.736 | 0.294 | 0.629 |
| Average | 98.91% | 80.61% | 99.02% | 97.08% | 0.386 | 0.312 | 0.188 | 0.229 |
| (Std) | (3.06%) | (17.09%) | (2.16%) | (2.79%) | (0.165) | (0.197) | (0.038) | (0.131) |

**Table 3: Comparison between the effectiveness on ChatGPT's responses by baseline prompts and APT-Pipe prompts.**

| | F1-score | | | Precision | | | Recall | | |
|---|---|---|---|---|---|---|---|---|---|
| | APT-Pipe | w/o Step 2 | w/o Step 3 | APT-Pipe | w/o Step 2 | w/o Step 3 | APT-Pipe | w/o Step 2 | w/o Step 3 |
| Ag's News | 88.32% | 83.43% | **88.74%** | 88.72% | 85.70% | **88.84%** | 88.31% | 83.87% | **88.76%** |
| SemEval-19 | 80.29% | - | 77.47% | 81.08% | - | 78.00% | 80.00% | - | 77.27% |
| Sarcasm | 55.31% | - | 51.54% | 66.88% | - | 69.28% | 59.87% | - | 58.38% |
| Clickbait | 94.56% | - | 93.90% | 94.64% | - | 94.25% | 94.57% | - | 93.92% |
| SemEval-16 | 50.93% | 34.90% | 45.10% | 57.95% | 52.37% | 53.43% | 47.28% | 31.33% | 41.53% |
| P-Stance | 74.82% | - | - | 77.27% | - | - | 75.33% | - | - |
| Vax-Stance | 66.25% | - | 63.11% | 69.97% | - | 65.50% | 64.91% | - | 62.81% |
| RU-Stance | 74.89% | - | 73.17% | 81.58% | - | 80.98% | 74.55% | - | 72.79% |
| Tweepfake | 75.27% | 62.40% | - | 76.20% | 70.81% | - | 75.46% | 64.86% | - |
| FakeHate | 49.51% | 24.53% | - | 74.56% | 67.35% | - | 49.18% | 31.54% | - |
| GPT-Article | 40.96% | 51.79% | 42.75% | 48.92% | 58.70% | 43.15% | 49.75% | 56.00% | 43.13% |
| GPT-Wiki | 65.11% | 39.54% | - | 67.17% | 67.13% | - | 64.15% | 48.33% | - |

**Table 4: Ablation study on prompt-tuning. The "*w/o* Step 2/3" represents the variant of APT-Pipe without prompt-tuning in Step 2/3. A "-" denotes that the ablation comparison is not available as APT-Pipe prompts on the corresponding dataset don't apply few-shot learning in Step 2 or no NLP metrics are selected in Step 3. The bold percentage denotes the highest value of corresponding metrics.**

| | Zero-shot | Few-shot |
|---|---|---|
| CoT | "Thought": "Let's think step by step." | "Thought": "Let's think step by step.", "Examples for thought": [{"Text": "{exemplar 1}", "Label": "{exemplar 1's ground-truth label}", "Explanation": "{CoT explanation for exemplar 1's ground-truth label}"},...] |
| ToT | "Thought": "Imagine three different experts are answering this question. All experts will write down 1 step of their thinking, then share it with the group. Then all experts will go on to the next step, etc. If any expert realises they're wrong at any point then they leave. Finally, all experts vote and elect the majority label as the final result." | "Thought": "Imagine three different experts are answering this question. All experts will write down 1 step of their thinking, then share it with the group. Then all experts will go on to the next step, etc. If any expert realises they're wrong at any point then they leave. Finally, all experts vote and elect the majority label as the final result.", "Examples for thought": [{"Text": "{exemplar 1}", "Label": "{exemplar 1's ground-truth label}", "Explanation": "{ToT explanation for exemplar 1's ground-truth label}"}, ...] |

**Table 5: A summary of key-value pairs augmented in prompt text in APT-Pipe prompts for applying Chain-of-Thought (CoT) and Tree-of-Thought (ToT).**

shows the outputs of Step 1 in the JSON column. Thus, we set up two additional variant pipelines, and compare them against APT-Pipe:

- `Without Step 2:` In this variant pipeline, we discard Step 2 from APT-Pipe. We directly apply the NLP feature selection module

(Step 3) to the initialized JSON prompt (from Step 1), and then add the selected NLP features to the encoded prompt.

- `Without Step 3:` In this variant pipeline, we remove step 3 from APT-Pipe. It only uses the few shots or the zero shots prompting mechanism from Step 2.

Table 4 compares ChatGPT's performance on the test sets using the original pipeline, and the two variants. APT-Pipe outperforms the variants in *nine out of twelve* datasets in terms of F1 scores. We observe that APT-Pipe outperforms the `Without Step 2` method by 6.12% on F1 score, 0.96% on precision, and 4.85% on recall. For the `Without Step 3` method, APT-Pipeline outperforms it by 1.31% on F1 score, 1.36% on precision, and 1.72% on recall. This demonstrate the effectiveness of our fully integrated prompting steps in improving the performance of ChatGPT for text annotation tasks.

## 6 EXTENDING APT-PIPE PROMPTS

APT-Pipe is built in a modular fashion and is extensible by design. To demonstrate this, we briefly conduct a case-study by extending APT-Pipe to support two additional state-of-art tuning mechanisms:

| | F1-score | | | Precision | | | Recall | | |
|---|---|---|---|---|---|---|---|---|---|
| | *w/i* CoT | *w/i* ToT | APT-Pipe | *w/i* CoT | *w/i* ToT | APT-Pipe | *w/i* CoT | *w/i* ToT | APT-Pipe |
| Ag's News | 84.26% | 85.60% | **88.32%** | 86.75% | 86.90% | **88.72%** | 84.59% | 85.79% | **88.31%** |
| SemEval-19 | 77.51% | 80.59%↑ | 80.29% | 78.16% | 80.79% | **81.08%** | 77.24% | 80.47%↑ | 80.00% |
| Sarcasm | 56.60%↑ | 57.56%↑ | 55.31% | **68.19%**↑ | 68.11%↑ | 66.88% | 60.98%↑ | 60.82%↑ | 59.87% |
| Clickbait | **95.43%**↑ | 92.71% | 95.35% | **95.52%**↑ | 93.42% | 95.39% | **95.43%**↑ | 92.73% | 95.34% |
| SemEval-16 | 45.08% | 45.42% | **50.93%** | 53.21% | 53.16% | **57.95%** | 42.14% | 42.64% | **47.28%** |
| P-Stance | 76.81% | 80.52%↑ | 77.86% | 79.95% | **82.22%**↑ | 81.02% | 77.43% | 80.78%↑ | 78.41% |
| Vax-Stance | 68.04%↑ | 59.50% | 65.58% | **71.54%**↑ | 60.25% | 69.31% | **67.00%**↑ | 58.96% | 64.26% |
| RU-Stance | 74.27% | 76.25%↑ | 74.89% | **81.97%**↑ | 81.12% | 81.58% | 73.87% | **75.66%**↑ | 74.55% |
| Tweepfake | 72.52% | 71.40% | **75.27%** | 73.32% | 74.68% | **76.20%** | 72.68% | 72.05% | **75.46%** |
| FakeHate | 49.18% | 20.99% | **49.51%** | 71.61% | 71.43% | **74.56%** | 50.41%↑ | 30.29% | 49.18% |
| GPT-Article | **55.04%**↑ | 49.86% | 40.96% | **55.43%**↑ | 50.06% | 48.92% | **54.97%**↑ | 50.31%↑ | 49.75% |
| GPT-Wiki | 67.23% | **70.58%**↑ | 65.11% | 69.41%↑ | **76.31%**↑ | 67.17% | 66.67%↑ | **69.81%**↑ | 64.15% |

**Table 6: Comparison between ChatGPT's performance within APT-Pipe prompts extended with CoT and ToT. The "*w/i* CoT (ToT)" represents the group of prompts by APT-Pipe extended with CoT (ToT). A "↑" beside a percentage means CoT/ToT improves APT-Pipe's performance. The bold percentage denotes the highest value of corresponding metrics.**

- **Chain-of-Thought (CoT):** CoT is a method to improve LLM's decisions by leading it to explain its behaviors or exemplars step by step [54].
- **Tree-of-Thought (ToT):** ToT decomposes prompt tasks in several steps and each step is reasoned over using multiple CoT explanations. The goal of ToT is to search a route on these CoT explanations step by step and output explanations to explain the LLM's behaviors [57].

***Implementation:*** For this experiment, we "plug-in" these two modules to create a fourth stage in the pipeline. Here, we embed the two proposed augmentations into the prompts output by APT-Pipe in Step 3 [27, 30]. Both augmentations are formed by adding instructive sentences that can be directly injected into prompts to guide ChatGPT's reasoning process. Table 5 lists the specific text for the two prompt additions. For CoT, its augmentation involves the simple addition of the instruction "*Let's think step by step.*" [30]. For ToT, its augmentation involves more comprehensive guidance as "*Imagine three different experts are answering this question. All experts will write down 1 step of their thinking, then share it with the group. Then all experts will go on to the next step, etc. If any expert realises they're wrong at any point then they leave. Finally, all experts vote and elect the majority label as the final result.*" [27].

Note that prompts tuned by APT-Pipe may employ either zero-shot or few-shot configurations. We refer to suggestions by relevant studies to set up the two types of automatic realizations of CoT/ToT under zero-shot and few-shot settings, respectively [36, 52]. For APT-Prompts with a zero-shot configuration, applying CoT and ToT simply adds their instructive sentences into the prompts (see Table 5). For APT-Pipe prompts with the few-shot configuration, for each prompt and its exemplars selected in Step 2, we ask ChatGPT to explain the exemplars' ground-truth labels through a prompt enhanced by CoT/ToT. Afterwards, we append these explanations to the exemplars in prompts correspondingly. The CoT/ToT prompt enhancement is as follows:

```
{
    "Prompt": "Follow the thought to reason the true label of
    following text among given labels for specified task.",
    "Thought": "{CoT/ToT instructive sentences}",
    "Text": "{text to classify}",
    "Task": "{task domain}",
    "True label": "{ground-truth label for text}",
    "Labels": ["Label 1", "Label 2", ...],
    "Desired format": {
        "Explanation": "<explanation_for_the_true_label>"
    }
}
```

***Results:*** Table 6 presents the performance on the test sets using the baseline APT-Pipe prompts (from Section 3) against their extended versions with CoT and ToT. CoT improves APT-Pipe's F1-scores on *five* datasets by 4.01% ($SD = 5.71\%$) on average, and ToT improves APT-Pipe's F1-scores on *six* datasets by 3.49% ($SD = 3.17\%$) on average. These results show that extending APT-Pipe with CoT and ToT has the potential to improve performance. However, we also identify some trade-offs for such extensions. For example, ToT decreases ChatGPT's precision on *eight* datasets by 3.84% ($SD = 3.16\%$) on average. This implies that extending APT-Pipe with ToT can make ChatGPT less likely to respond with precise annotations and introduce more false positive answers instead. Importantly, APT-Pipe can evaluate the efficacy for any given dataset and only include the prompt additions that result in performance improvements. We, therefore, show this to highlight APT-Pipe's modular flexibility.

## 7 CONCLUSION & DISCUSSION

***Summary:*** This paper has proposed APT-Pipe, an automatic prompt-tuning tool for annotating social computing data. Given a text classification dataset, APT-Pipe aims to automatically tune a prompt for ChatGPT to reproduce identical annotations to a small set of ground-truth labels. APT-Pipe then tests a variety of these prompts to identify the one that performs best. Once complete, the tuned prompt can be applied across the full dataset. Our results show that APT-Pipe can improve ChatGPT's overall performance on nine datsets, improving its F1-score by 7.01% on average. Importantly, in cases where the pipeline's prompt tuning techniques do not work well for the specific data, APT-Pipe makes this transparent, allowing the engineer to use the un-tuned prompt instead.

***Limitations & Future work:*** As the first attempt at automating prompt-tuning for data annotation, our study has certain limitations. These form the basis of our future work. *First*, we only test APT-Pipe with a small number of datasets from three task domains. We are keen to understand how APT-Pipe performs on other classification domains like hate speech detection [26] and relation extraction [24]. *Second*, in our evaluation, we assume that the prompt engineer can annotate small parts of their datasets, to underpin APT-Pipe. Typically, this involves labeling hundreds of posts. We wish to further investigate APT-Pipe's performance on far fewer labels (*e.g.,* 10 samples), and explore techniques to minimize human burden. *Third*, as an extensible framework, APT-Pipe has only been evaluated with two modules — few-shot learning and NLP metrics selection — alongside the CoT and ToT extensions. We conjecture that numerous new prompting tuning techniques will be proposed in the coming years. We are therefore excited to integrate future tuning techniques into the pipeline. As such, APT-Pipe is open-source and will be made publicly available.

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

## A ETHICAL CONSIDERATIONS

Our study is based on analyzing publicly released text classification datasets. We access to data from published literature and public codebase sharing platform involving *GitHub*, *Hugging Face*, and *Kaggle*. Our target to utilize these datasets is only to assess APT-Pips's effectiveness on improving ChatGPT's annotation performance. Our analyses all follow the data policies of above platforms.

## B SUMMARY OF DATASETS

| | Dataset | Size | Description | Label |
|---|---|---|---|---|
| **News Classification** | Ag's News | *3,000 | A dataset for categorizing news articles' themes [61]. | World, Business, Sports, Sci/Tech |
| | SemEval-19 | 639 | A dataset for classifying whether news articles are hyperpartisan [29]. | (non-) hyperpartisan |
| | Sarcasm | *3,000 | A dataset for detecting sarcasm in news headlines [42]. | (not) sarcastic headline |
| | Clickbait | *3,000 | A dataset for classifying whether news headlines are clickbaits [8]. | (not) clickbait headline |
| **Stance Detection** | SemEval-16 | *3,000 | A dataset for classifying tweets' stances towards five controversial topics (*e.g.*, legalization of abortion, feminist movement, Atheism, etc.) [43]. | favor/neutral/against-{topic} (*e.g.*, favor-Atheism) |
| | P-Stance | *3,000 | A dataset for classifying tweets' stances towards three politicians [34]. | favor/against-{politician} (*e.g.*, favor-Joe Biden) |
| | Vax-Stance | 1,604 | A dataset for classifying tweets' stances towards COVID-19 vaccine [47]. | Pro/Anti-Vaccine, Neutral |
| | RU-Stance | 1,460 | A dataset for classifying tweets' stances on the Russo-Ukrainian War [64]. | pro-Russia/Ukraine |
| **AI-writing Detection** | Tweepfake | *3,000 | A dataset for detecting whether tweets are generated by AI [15]. | ai, human |
| | FakeHate | *3,000 | A dataset for detecting whether hate speeches are generated by AI [12, 22]. | ai, human |
| | GPT-Article | 1,018 | A dataset for detecting whether textual articles are generated by GPT [39] | ai, human |
| | GPT-Wiki | 306 | A dataset for detecting whether Wiki articles are generated by GPT [23] | ai, human |

**Table 7: A summary of datasets.**

## C ALGORITHM FOR STEP 3

---

**Algorithm 1:** Algorithm for Step 3

---

**Input:** $D_{train} = (\mathbb{T}_{train}, \mathbb{L}_{train}, \mathbb{E}_{train})$: Training Dataset,
$\quad D_{val} = (\mathbb{T}_{val}, \mathbb{L}_{val}, \mathbb{E}_{val})$: Validation Dataset,
$\quad \mathbb{T}$ for text data, $\mathbb{L}$ for label, $\mathbb{E}$ for NLP metrics embeddings
$\quad \mathbb{P}_{val}$: Step 2 Prompts for validation dataset,
$\quad M_{nlp} = \{sentiment, emotion, toxicity, topic\}$: NLP metrics,
$\quad N$: The number of NLP metrics,
$\quad l_T$: the size of $D_{train}$, $l_V$: the size of $D_{val}$,
$\quad ChatGPT()$: ChatGPT API Calling Function.
**Output:** Ranked list of selected NLP metrics $[M_1, M_2, ...]$

1 $M_{select} = [];$
2 $M_{cand} = M_{nlp};$
3 $\mathbb{P}_{val}^{iter\ 0} = \mathbb{P}_{val};$
$\quad$ /* $\mathbb{R}$ denotes ChatGPT's response $\quad$ */
4 $\mathbb{R}_{val}^{iter\ 0} = ChatGPT(\mathbb{P}_{val}^{iter\ 0});$
5 $F1_{val}^{iter\ 0} = \text{F1\_Score}(\mathbb{R}_{val}^{iter\ 0}, \mathbb{L}_{val});$
6 **for** $i \leftarrow 1$ **to** $N$ **do**
$\quad$ /* Rank the candidate NLP metrics based on the XGBoost accuracy $\quad$ */
7 $\quad$ **for** $j \leftarrow 0$ **to** $Size(M_{cand}) - 1$ **do**
8 $\quad\quad X = concat([One\_Hot(\mathbb{L}_{train}), \mathbb{E}_{train}[M_{cand}[j]]]);$
9 $\quad\quad Y = [\mathbb{R}_{train} == \mathbb{L}_{train}];$
10 $\quad\quad XGB_{accuracy}[M_{cand}[j]] = XGBoost(X, Y);$
11 $\quad$ **end**
12 $\quad M_{rank} = \text{rank } M_{cand} \text{ based on } XGB_{accuracy};$
$\quad$ /* Sequentially Search the ranked NLP metrics to the first one that increases the F1 score compared with the former iteration */
13 $\quad S_{cand} = Size(M_{cand});$
14 $\quad$ **for** $j \leftarrow 0$ **to** $Size(M_{rank}) - 1$ **do**
$\quad\quad$ /* Add current NLP prompt into the prompt in the former iteration $\quad$ */
15 $\quad\quad \mathbb{P}_{val}^{iter\ i} = \text{Add } M_{rank}[j] \text{ metrics in } \mathbb{P}_{val}^{iter\ (i-1)};$
16 $\quad\quad \mathbb{R}_{val}^{iter\ i} = ChatGPT(\mathbb{P}_{val}^{iter\ i});$
17 $\quad\quad F1_{val}^{iter\ i} = \text{F1\_Score}(\mathbb{R}_{val}^{iter\ i}, \mathbb{L}_{val});$
18 $\quad\quad$ **if** $F1_{val}^{iter\ i} > F1_{val}^{iter\ (i-1)}$ **then**
19 $\quad\quad\quad M_{cand} = M_{cand} - M_{rank}[j];$
20 $\quad\quad\quad M_{select} = M_{select} + M_{rank}[j];$
21 $\quad\quad\quad$ break;
22 $\quad\quad$ **end**
23 $\quad$ **end**
$\quad$ /* If no beneficial NLP metric found, return the existing selected NLP metrics $\quad$ */
24 $\quad$ **if** $Size(M_{cand}) == S_{cand}$ **then**
25 $\quad\quad$ break;
26 $\quad$ **end**
27 **end**
28 **return** $M_{select}$

---

