# OpenReview forum: "APT-Pipe: An Automatic Prompt-Tuning Tool for Social Computing Data Annotation"
_ACM.org/TheWebConf/2024/Conference — TheWebConf24 Oral_

### Official Review · Reviewer_Mswz · 2023-11-16

**Novelty:** 4
**Technical Quality:** 4

**Review:**

1. The paper introduces APT-Pipe, an automatic and extensible prompt-tuning pipeline. This tool is designed to enhance the performance of ChatGPT in social computing data annotation tasks. It represents a significant step forward in automating and improving the efficiency of data annotation using LLMs.
2. APT-Pipe has been shown to improve ChatGPT's classification performance across a variety of datasets. Specifically, it enhances ChatGPT's F1-score by an average of 7.01% on nine out of twelve datasets tested. This improvement is a testament to the effectiveness of APT-Pipe in refining the accuracy of LLMs in classification tasks.
3. The prompts generated by APT-Pipe not only produce annotations in a consistent format but also reduce the time required for classification tasks. Over 97% of responses generated by APT-Pipe are in a parsable format, and it reduces the time cost by more than 25% compared to baseline methods. This aspect is crucial for large-scale data annotation projects where time efficiency and consistency are paramount.
4. The paper highlights the extensibility of APT-Pipe by incorporating two additional prompt-tuning techniques: Chain of Thought (CoT) and Tree of Thought (ToT). These techniques further improve the F1-scores on various datasets, demonstrating the flexibility and adaptability of APT-Pipe to integrate new and emerging prompt-tuning methods.
5. APT-Pipe's ability to automate and enhance the data annotation process has significant implications for social computing research. It addresses the challenges of time consumption and cost associated with human-annotated datasets, offering a robust alternative that leverages the capabilities of LLMs.

**Questions:**

1. How well does APT-Pipe generalize across different types of social computing tasks beyond the ones tested? Can it adapt effectively to new, unseen datasets or tasks that significantly differ from the ones used in your experiments?
2. How does the performance of APT-Pipe compare with other state-of-the-art prompt-tuning or annotation tools? Are there specific scenarios where APT-Pipe outperforms these tools, or vice versa?
3. Given the importance of prompt design in the performance of ChatGPT, how does APT-Pipe ensure the optimal selection of prompts? What strategies are employed to handle the variability and complexity of natural language in prompt design?
4. The paper mentions the use of additional metrics like sentiment, emotion, and toxicity in prompt augmentation. How do these metrics specifically contribute to the performance improvement, and are there any limitations or challenges associated with their integration?
5. What kind of user interface does APT-Pipe offer, and how accessible is it for researchers or practitioners with varying levels of expertise in prompt-tuning or large language models?
6. Are there any ethical considerations or potential biases that need to be addressed when using APT-Pipe for social computing data annotation, especially given the sensitive nature of some social computing tasks?

**Reviewer Confidence:**

3: The reviewer is confident but not certain that the evaluation is correct

**Scope:**

3: The work is somewhat relevant to the Web and to the track, and is of narrow interest to a sub-community

---

### Official Review · Reviewer_thJ4 · 2023-11-18

**Novelty:** 6
**Technical Quality:** 5

**Review:**

The paper proposes APT-Pipe, an automatic prompt-tuning tool for data anotation. The proposed technique, APT-Pipe uses a three-step pipeline that generates JSON template prompts, selects few-shot examples, and augments prompts with NLP metrics. The paper evaluates APT-Pipe on twelve datasets from three domains and shows that it can enhance ChatGPT's performance on nine datasets, with an average improvement of 7.01% F1-score. The paper also demonstrates APT-Pipe's extensibility by integrating two additional tuning techniques, Chain-of-Thought and Tree-of-Thought.

Strengths
+ The paper addresses an important problem of automating prompt-tuning for data annotation tasks.
+ The paper presents a clear and detailed description of the proposed pipeline and its components. The paper provides a comprehensive evaluation of APT-Pipe on various datasets and domains, and compares it with several baselines. The paper also shows the flexibility of APT-Pipe by extending it with two state-of-the-art tuning methods.

Weaknesses
- The paper only tests APT-Pipe with one LLM, ChatGPT, and does not explore how it generalizes to other mdoels.
- The paper assumes that the prompt engineer can annotate a small subset of data, which may not be feasible or desirable in some scenarios.
- The paper does not provide any qualitative analysis or examples of the generated prompts and annotations.

**Questions:**

Please see the weaknesses pointed out in the review

**Reviewer Confidence:**

2: The reviewer is willing to defend the evaluation, but it is likely that the reviewer did not understand parts of the paper

**Scope:**

3: The work is somewhat relevant to the Web and to the track, and is of narrow interest to a sub-community

---

### Official Review · Reviewer_AU7H · 2023-11-21

**Novelty:** 4
**Technical Quality:** 5

**Review:**

**Quality**

This paper presents an automated prompt construction framework named APT-Pipe, which aims to automatically adjust prompts to improve the performance of text-based annotation tasks. The authors thoroughly explain how this framework addresses three key challenges and propose a three-stage prompt construction process. This technique can reduce the manual labor cost of building prompts by hand. However, it still requires humans to manually label a small dataset before constructing prompts automatically. The experimental section of the paper does not investigate the impact of the size of this dataset on the performance of the automatically constructed prompts. If a large annotated dataset is needed, this contradicts the paper's mentioned advantage of reducing manual labor costs.

**Clarity**

The paper is well-described, and the sentences are clear without ambiguity.

**Originality**

1. The authors introduce an automated prompt construction technique named APT-Pipe, which can enhance the performance of ChatGPT in social computation data annotation tasks.
2. The effectiveness of APT-Pipe was validated across twelve different text classification datasets. Results show that APT-Pipe can improve ChatGPT's classification performance on nine of these datasets, with an average increase in F1 score of 7.01%.
3. Prompts generated by APT-Pipe produce consistent annotations and require less time compared to existing prompt designs.
4. The authors demonstrate the scalability of APT-Pipe by incorporating two latest prompt adjustment techniques: Chain of Thought (CoT) and Tree of Thought (ToT). APT-Pipe can learn and automatically adjust prompts based on each dataset, with these enhancements, without the need for manual intervention.

**Significance**

Prompt-based LLM (Large Language Model) applications can significantly improve cost and speed for researchers. However, the performance of such models is heavily influenced by the quality of the prompts; poorly adjusted prompts can greatly diminish the model's performance on specific tasks. Therefore, research into how to construct effective prompts is meaningful.

**Cons**

* The method mentioned in Section 1 states that it "only requires humans to annotate a small sample subset of data to offer a robust ground truth". However, in practice, the performance of this "small sample" in experiments is not necessarily "small". That is to say, prompt engineers still need to annotate a considerable amount of data.
* This method assumes that prompt engineers can annotate the dataset, but this requires them to possess a certain level of domain knowledge. However, not every prompt engineer possesses this capability.
* The number of labels in this experimental dataset is relatively low, and this might be a reason why this method works. However, as the number of labels in the dataset increases, the amount of data that prompt engineers need to annotate may need to significantly increase as well. Moreover, the quality of annotation may greatly impact the subsequent results of text classification.
* This method is experimented upon based on ChatGPT, but utilizing ChatGPT multiple times for such an approach incurs a considerable cost to some extent.

**Questions:**

1. In the Step 2 of prompt construction, a Text Embedding Model is used to compute embeddings of texts in the dataset to determine their similarity to the text to be classified. The most similar samples are then chosen as examples to be included in the prompt. It's worth questioning what performance could be achieved if this model were directly used for classification. Is there a possibility that it might yield better results compared to using a larger model?
2. In the Step 3 of prompt construction, how is the metric for each sample calculated? Similarly, if the metric is directly used for predicting data, what level of performance could be achieved?
3. Has there been any testing on approximately how much data a prompt engineer needs to annotate to make this method reach an effective level? Because based on Appendix B, it seems that the required amount of annotation may not be insignificant.
4. The dataset used in the experiment has a relatively small number of labels, only a few. If the dataset has a larger number of labels, would this method still be effective? Would there be a significant increase in the amount of data that prompt engineers need to annotate?
5. In Step 2, there is a need for a text embedding model to embed the text, and based on this result, the text is sorted. Then, the top k texts and their labels are selected based on text similarity. Does this approach essentially inform ChatGPT in advance about the approximate range of labels to be selected, thereby enhancing the performance of ChatGPT?
6. In the experimental section, the method was also compared with different forms of regular prompts in time cost. Is this related to the ChatGPT server and not necessarily attributable to the merit of this method?
7. Is informing ChatGPT about the metrics to be used in Step 3 similar to a scenario where a judge pre-informs you about the evaluation criteria they will use to judge a case?

**Reviewer Confidence:**

4: The reviewer is certain that the evaluation is correct and very familiar with the relevant literature

**Scope:**

3: The work is somewhat relevant to the Web and to the track, and is of narrow interest to a sub-community

---

### Official Review · Reviewer_8RSz · 2023-11-23

**Novelty:** 6
**Technical Quality:** 5

**Review:**

This paper presents APT-Pipe, a tool that automatically tunes prompts to improve ChatGPT’s text classification performance. The author evaluated their tool on twelve text classification datasets and showed that it achieves an average improvement of 7% in weighted F1-score. Additionally, they highlight the tool’s versatility by showcasing its adaptability for accommodating supplementary tuning mechanisms.

Strengths:

* S1: The paper is well written, engaging, and addresses a tangible challenge and presents an opportunity to enhance labeling methods, improving LLMs' text classification performance.

* S2: I appreciate APT-Pipe's modular and extendable structure, showcased by the authors through the integration of two state-of-the-art tuning mechanisms. This ability to add or modify components significantly sets apart the innovation in this field.

Weaknesses:

* W1: Considering that "*the improvements in ChatGPT’s performance vary across these three tasks*," I'd value it if the authors expanded APT-Pipe's testing scope to include datasets beyond just these three task domains.

* W2: While not a direct flaw, utilizing a domain expert annotator, even alongside APT-Pipe, is preferable for generating the initial sample. Thanks to APT-Pipe, this sample could potentially be smaller in size.

Please note that I outlined both weaknesses prior to reaching the limitations section. I appreciate the authors' acknowledgment of these limitations.

**Questions:**

* Q1: "*Considering the existing literature, it's apparent that current prompt-tuning methods rely on domain expertise and human judgment, limiting the use of LLMs in automated annotation tasks.*" Even though APT-Pipe notably reduces annotators' efforts, there's still a requirement for domain expertise to significantly enhance classification, Were the authors able to test APT-Pipe using annotations from individuals who were not domain experts?

* Q2: A small note: I got a bit confused while going through subsection 3.3 and examining figure 3 regarding the origin of the training and validation datasets—who generated them: prompt engineers or ChatGPT—and at which step.

* Q3: APT-Pipe relies on various elements, such as metrics and the classifier for each NLP metric. I believe altering any of these components could yield diverse results.Did the authors conduct experiments using various models and metrics?

* Q4: What, in the authors' opinion, could lead to a boost exceeding 7% in improvement?

**Reviewer Confidence:**

3: The reviewer is confident but not certain that the evaluation is correct

**Scope:**

4: The work is relevant to the Web and to the track, and is of broad interest to the community

---

### Official Review · Reviewer_HeHY · 2023-11-29

**Novelty:** 2
**Technical Quality:** 4

**Review:**

Summary: The paper proposes a prompt-tuning approach that leverages a manually annotated dataset to automatically create prompts that can improve data annotation performance of GPT.

Strengths:

1.	The paper proposes an end-to-end approach.

2.	The proposed framework is flexible, and it can accommodate contemporary prompt-tuning approaches.

3.	Evaluation was done on multiple publicly-available datasets from three different domains.

4.	Carefully crafted illustrations improve the comprehensibility of the contributions.

Weaknesses:

1.	The framing of the problem (as I understood from Introduction and Related Work) is a bit confusing. Why would one use GPT for creating annotated datasets with the intention of training future classification models, if GPT itself can annotate data accurately?

2.	The Related Work and Evaluation exclude a whole body of prior work on social media data classification before the GPT/ChatGPT era. There have been plenty of research works on detecting hate speech, analyzing sentiments, detecting fake news, etc., that do not rely on GPT for classification. How does the proposed APT-Pipe compare to these works with regard to the classification performance on publicly-available datasets?

3.	If the method requires manual construction of a ground-truth dataset anyway, why not just fine tune LLMs instead of creating prompts in a round-about away? In this case, it is also important to compare performances of prompt-driven annotation and fine tuning-based annotation.

4.	The techniques in the three steps of APT-Pipe are relatively simple with limited novelty. Specifically, Step 1 involves manually engineering of a prompt template, step 2 does cosine similarity-based retrieval of samples, and step 3 implements a simple feature selection method – a topic that has been well researched in classical machine learning.

5.	The proposed research is not reproducible as of now since the code is not publicly available. However, the authors have indicated in the paper that they will make it available.

6.	The choice of baselines seems to a little biased towards APT-Pipe. Intuitively, one can expect that APT-Pipe with all the extra contextual information, to do well against zero-shot basic-template alternatives. It is not clear if the baselines were also strengthened with few-shot ground-truth examples.

**Questions:**

Please refer to the Weaknesses in the review. Other than the questions there, I have the following questions:

1.	What is the difference between contributions 2 and 3 in the Introduction section?

2.	In Line 216 on page 2, what is the “remaining dataset”?

3.	In Time Cost part of section 5.2, it is first stated that APT-Pipe decreases time cost compared to JSON prompts, and in the very next sentence, it is stated that APT-Pipe introduces a small additional cost compared to JSON prompts. Which one is true?

**Reviewer Confidence:**

3: The reviewer is confident but not certain that the evaluation is correct

**Scope:**

2: The connection to the Web is incidental, e.g., use of Web data or API

---

### Official Review · Reviewer_sDyp · 2023-12-01

**Novelty:** 5
**Technical Quality:** 4

**Review:**

This paper presents a APT-Pipe, an extensible automatic prompt tuning tool for annotating social computing data. APT-Pipe automatically tunes prompts to enhance ChatGPT's text classification performance. APT-Pipe is evaluated on 12 datasets from 3 domains. The evaluation results show that APT-Pipe improves ChatGPT's performance on 9 datasets, achieves higher F1-score, and takes less time.

Strengths
+ Well written paper that explains each step systematically; adds to reproducibility
+ Extensive evaluation on 12 datasets from 3 different domains
+ Detailed ablation study
+ Extensible tool; extensions demonstrated via two case studies

Weakness
- Whereas the paper describes "what was done" effectively and extensively, it does not offer much explanation on the rationales or "why" for the choices made. For instance, paper leverages [56] for identifying text samples for manual annotation. However, does not provide details about why. Similarly, the paper mentions that the tool considers NLP metrics of sentiment, emotion, toxicity, and topic. However, there is no explanation on why these metrics are appropriate metrics to be considered. Further 12 datasets that are selected for evaluation have no justification. The paper will benefit from including rationale for choices made: ChatGPT settings, dataset split, etc.
- The paper needs to better motivate its relevance to the Web conference. Why is the paper better suited here and not in an NLP or human computation conference? All but one reference is from NLP or human computation.
- (minor) Tables 3--6 have been compressed to fit a column and save space. As a result, text is very small to be read on a printed paper.

Suggestion
- Because this is a human-annotation task, an alternative metric could be the number of annotations other baselines require to achieve the same performance or F1-score.

**Questions:**

- Why Yang et al. [56] to select samples for manual annotation? Are there other alternatives?
- Why sentiment, emotion, toxicity, and topic as NLP metrics? Are these metrics equally sufficient and necessary for the 12 datasets being considered for evaluation?
- I couldn't completely understand parsability. Could you please explain it a little more with an example?
- Could you please motivate the paper's contributions' relevance to the Web conference?

**Ethics Review Description:**

-

**Reviewer Confidence:**

1: The reviewer's evaluation is an educated guess

**Scope:**

2: The connection to the Web is incidental, e.g., use of Web data or API

---

### Decision · Program_Chairs · 2024-01-22

**Decision:**

Accept (Oral)

**Comment:**

Summary: The paper looks into an extensible automatic prompt tuning tool for annotating social computing data.

 Strengths:
 + well-written, systematic description
 + extensive experimental evaluation icluding ablation
 + extensible applicability, illustrated via case studies

 Weaknesses:
 - need for more explanations on the design choices
 - some concern about relevance to Web
 - some discussion needed on the costs of reliance on ChatGPT for the study

 Recommendation: Accept. Empirical work of recent interest.